# Dual-Stage Deeply Supervised Attention-Based Convolutional Neural Networks for Mandibular Canal Segmentation in CBCT Scans

**DOI:** 10.3390/s22249877

**Published:** 2022-12-15

**Authors:** Muhammad Usman, Azka Rehman, Amal Muhammad Saleem, Rabeea Jawaid, Shi-Sub Byon, Sung-Hyun Kim, Byoung-Dai Lee, Min-Suk Heo, Yeong-Gil Shin

**Affiliations:** 1Center for Artificial Intelligence in Medicine and Imaging, HealthHub Co., Ltd., Seoul 06524, Republic of Korea; 2Department of Computer Science and Engineering, Seoul National University, 1 Gwanak-ro, Gwanak-gu, Seoul 08826, Republic of Korea; 3Division of AI and Computer Engineering, Kyonggi University, Suwon 16227, Republic of Korea; 4Department of Oral and Maxillofacial Radiology, School of Dentistry, Seoul National University, 1 Gwanak-ro, Gwanak-gu, Seoul 08826, Republic of Korea

**Keywords:** mandibular canal, 3D segmentation, jaw localization, CBCT

## Abstract

Accurate segmentation of mandibular canals in lower jaws is important in dental implantology. Medical experts manually determine the implant position and dimensions from 3D CT images to avoid damaging the mandibular nerve inside the canal. In this paper, we propose a novel dual-stage deep learning-based scheme for the automatic segmentation of the mandibular canal. In particular, we first enhance the CBCT scans by employing the novel histogram-based dynamic windowing scheme, which improves the visibility of mandibular canals. After enhancement, we designed 3D deeply supervised attention UNet architecture for localizing the Volumes Of Interest (VOIs), which contain the mandibular canals (i.e., left and right canals). Finally, we employed the Multi-Scale input Residual UNet (MSiR-UNet) architecture to segment the mandibular canals using VOIs accurately. The proposed method has been rigorously evaluated on 500 and 15 CBCT scans from our dataset and from the public dataset, respectively. The results demonstrate that our technique improves the existing performance of mandibular canal segmentation to a clinically acceptable range. Moreover, it is robust against the types of CBCT scans in terms of field of view.

## 1. Introduction

The Inferior Alveolar Nerve (IAN), also known as the mandibular canal, is the most critical structure in the mandible region which supplies sensation to the lower teeth. Similarly, sensation to the lips and chin is provided by the mental nerve, which passes through the mental foramen [1]. An essential step in implant placement, third molar extraction and various other craniofacial procedures, such as orthognathic surgery, is determining the position of the mandibular canal. It is also crucial for diagnosing vascular and neurogenic diseases associated with the nerve, diagnosing lesions near the mandibular canal and planning oral and maxillofacial procedures. If the mandibular canal gets injured during any of these processes, patients may experience aches, pain and temporary paralysis [2,3]. Therefore, preoperative treatment planning and simulation are necessary to avoid nerve injury and the identification of the exact location of the mandibular canal aids in achieving the required planning strategy for the patient [4].

One of the most frequently used three-dimensional (3D) imaging modalities for preoperative treatment planning and postoperative assessment in dentistry is the Cone Beam Computed Tomography, also known as CBCT [5]. The CBCT volume is reconstructed using projection images realized from different angles with a cone-shaped beam and stored as a sequence of axial images [6]. Multi-Detector Computed Tomography (MDCT) is a clinical replacement for CBCT; however, high radiation doses and insufficient spatial resolution limit its application. In contrast, the CBCT allows more precise imaging of hard tissues in the dentomaxillofacial area and its effective radiation dosage is lower than that of the MDCT1. CBCT is also inexpensive and readily available. Nonetheless, in practice, there are certain challenges associated with mandibular canal segmentation from CBCT images, such as inaccurate density and large amount of noise [7].

Surgical planning and pre-surgical examination are crucial in dental clinics. A standard imaging tool used for such assessments and planning is the panoramic radiography, which is constructed from a dental arch to provide all the relevant information in a single view. These radiographs bear disadvantages, such as difficulty in determining the 3D rendering of an entire canal and connected nerves [8]. Another common preoperative assessment approach is annotating the canal in 3D images to produce the segmentation of the canal. This kind of manual annotation is very knowledge-intensive, time-consuming and tedious. Thus, there is a need for a tool to assist the radiologist and reduce the burden by using automatic or semi-automatic segmentation of the canal.

Complications in accurate segmentation of the mandibular canal arise as the CBCT values of the mandibular canal are similar to surrounding tissues in the mandible region. General parameters associated with imaging, i.e., scan resolution, pixel spacing and pixel values, also significantly influence the segmentation performance. Additionally, other characteristics of the mandibular canal, such as the curvature of the canal and the geometry, also impact the segmentation accuracy. In the current dentistry workflow, dentists usually use manual delineation or semi-automatic preoperative segmentation of the mandible. Manual delineation requires experienced dentists to use software to delineate the contour of the mandibular canal on each slice of the CBCT scan. Semi-automatic segmentation includes region growth, level set and other methods requiring continuous interactive operations. These methods make the segmentation process slow and inefficient [9,10], greatly increasing the workload of implant doctors. Therefore, improving the segmentation efficiency of the mandibular canal has become an urgent problem to be solved for implant planning software.

Several studies have attempted to overcome the challenges mentioned above by developing various systems for automatic segmentation of the mandibular canal in CBCT scans. Such systems include classical image processing-based techniques and advanced deep learning-based methods [11]. Classical methods mostly rely on raw voxel values and consider 2D contextual information to determine the mandibular canal position. These canal analyses lack the 3D sequential perspective, which limits their performance and robustness. On the other hand, DL-based methods have shown the potential to segment the 3D structures in various 3D imaging modalities accurately. However, DL-based techniques require extensive well-annotated data to develop an accurate and generalized algorithm. Although a significant amount of raw CBCT scans are available, obtaining annotation on 3D scans is a serious obstacle. Additionally, there is a lack of publicly available annotated data sets due to the privacy constraints associated with the sharing of medical data, owing to the patient’s personal information.

To overcome these challenges, in this study, we first develop the largest CBCT dataset, which consists of 1010 3D scans with mandibular canal annotation. Then, for automatic segmentation of the mandibular canal in 3D CBCT scans, we design a dual-stage 3D Convolutional Neural Network (CNN)-based technique. The proposed framework first localizes the mandibular region using naive segmentation produced by deeply supervised attention UNet. In the second stage, volumes of interest are extracted from a full 3D CBCT scan to apply multi-scale deeply supervised UNet architecture for mandibular canal segmentation. The proposed technique has been rigorously evaluated on 500 scans of our dataset and 15 scans from the publicly available dataset. The results demonstrate that our framework not only outperforms the existing methods in terms of dice score and mIoU but also exhibits significant robustness regarding scan variations.

The rest of the paper is organized as follows. In Section 2, we present related works. In Section 3, the details of each step of our proposed method as well as the materials used are described. In Section 4, we present the obtained results and comparison of our study with other studies as well as analysis and discussion of our work and then we conclude in Section 5.

## 2. Related Work

Several efforts have been made to develop semi-automated or fully automated solutions for automatic segmentation of mandibular canal [11]. Based on the techniques utilized for development, these systems can be classified into two categories, i.e., classical image processing-based methods and advanced deep learning-based techniques.

In [12], Kim et al. utilized 3D panoramic Volume Rendering (VR) and texture analysis techniques for mandibular canal segmentation. Specifically, they introduced color shading and compositing methods in 3D panoramic VR for the enhancement of foramens and later they employed line tracking to compute the path of mandibular canal. Similarly, in [13], Abdolali et al. presented a hybrid framework that combines anatomical and statistical information to segment the mandibular canal. They first applied a low-rank decomposition-based algorithm for pre-processing, which was later combined with a statistical shape model and fast marching to segment the mandibular bone and canals. The study reported an improved performance in terms of Average Symmetric Surface Distance (ASSD) and average mean curve distances. They extended their work in [14] by employing a Lie group-based statistical shape model to represent the shape variations and applied fast marching to localize the mandibular canal in CBCT scans. This extension achieved a significantly higher performance in terms of dice score and symmetric distance on their private dataset. In [15], Wei et al. first applied windowing and then K-mean clustering algorithm to cluster the texture parameters to improve the visibility of the mandibular canal in Multi-Plane Reconstruction (MPR) views. Finally, a 2D line-tracking method was applied for rough segmentation of the mandibular canal, further refined by fitting the fourth-order polynomial.

Although classical image processing-based studies have reported promising results on limited private datasets, these methods lack generalization ability, making them inefficient for real-time application. On the other hand, deep learning-based methods have made vast inroads into various computer-aided medical applications [16], such as disease detection [17] and the segmentation of affected regions [18]. Consequently, deep learning has also been applied for mandibular canal segmentation to boost performance.

For instance, Kwak et al. [19] implemented three deep learning models, i.e., 2D SegNet, 2D and 3D UNets, for mandibular canal segmentation. Prior to the segmentation, they applied thresholding-based teeth segmentation to eliminate the non-mandibular region from 3D CBCT scans. The study suggests that the 3D UNet outperforms both 2D models. In another study, Jaskari et al. [20] presented a 3D fully Convolutional Neural Network-based technique for mandibular canal segmentation. They evaluated their model on private data consisting of 15 scans and achieved dice scores of 0.57 and 0.58 for the left and right canals, respectively. Similarly, Faradhilla et al. [21] also presented a Residual Fully Convolutional Network (RFCN) with dual loss functions, i.e., non-mandibular region and boundary of mandibular canal-based loss functions, for segmenting the mandibular canal in 2D parasagittal views of CBCT scans. The study used 500 parasagittal 2D images for validation and reported promising results in terms of dice score.

Furthermore, Widiasri et al. [22] simultaneously detected alveolar bone and mandibular canal on 2D coronal views of CBCT scans by applying a modified version of YOLOv4 [23]. However, these methods can be classified as semi-automated as they require 2D views generated by manual inputs from dentists. Dhar et al. [24] used a model based on 3D UNet to segment the canal. They used pre-processing techniques to generate the center lines of the mandibular canals and used them as ground truths in the training process. Verhelst et al. [25] used a patch-based technique to localize the jaw and then applied the 3D UNet model to segment the canal in that ROI. In a similar way, Lahoud et al. [26] first coarsely segmented out the canal and then using the patches extracted based on this coarse segmentation, performed fine segmentation of the canal. Capriano et al. [27] developed a novel and large publicly available dataset for applying deep learning with dense and sparse annotations on CBCT scans. To generate dense voxel-level annotations, they reconstructed the polygon mesh in the form of the α-shape. They exploited their developed dataset to achieve state-of-the-art performance by employing a deep learning-based method proposed by Jaskari et al. [20]. The same authors extended their work in [28] by leveraging their dataset with 3D dense annotations to train a deep label propagation model which outperformed the previous techniques.

Alternatively, Du et al. [29] proposed another framework based on 3D Convolutional Neural Networks (CNNs) trained using the dataset developed by Capriano et al. [27]. In contrast to Capriano et al. [27], they first generated the annotations by employing the centerline combined with regional growth method. Afterward, they incorporated Spatial-Channel Squeeze and Excitation attention scheme to a 3D UNet architecture and achieved significantly better performance with respect to their annotation scheme. However, their results cannot be compared with Capriano et al. [27] as they used different target annotations to train their model.

Table 1 summarizes recently published deep learning-based studies for mandibular canal segmentation. It also presents the techniques used in each study with the nature of accessibility of datasets (i.e., private or public) along with the types of data used as input to the models, i.e., full and medium views. It can be observed that each listed study which achieved significantly higher performance utilized medium view, that is, the sub-volume of full view as described in Figure 1. In contrast, only a single study utilizing the full view, full-face 3D CBCT scan, was found. However, it obtained limited performance. In this work, we employed a dual-stage mechanism to automatically localize the mandibular canal region in full view, after which Volumes Of Interest (VOIs) are extracted to segment the mandibular canal. The proposed method has been trained and evaluated on the largest CBCT dataset, consisting of 1010 3D CBCT full-view scans. The results demonstrate that our framework outperforms state-of-the-art segmentation performance and offers better generalization ability.

## 3. Materials and Methods

### 3.1. Study Design

The objective of this study is to design a deep-learning approach for automatic mandibular canal segmentation. The study design consists of pre-processing, model training and post-processing, each discussed in detail in the sections below. The details about the dataset and pre-processing steps are explained in Section 3.2 and Section 3.3, respectively. The network design is discussed in detail in Section 3.4 Network Architecture. The network was validated on 500 scans.

### 3.2. Datasets

In this work, we utilized two datasets; one which we developed in this study and the second, a publicly available dataset released by Cipriano et al. [27]. The following subsections provide details about both datasets and finally we summarize their comparison in Table 2.

#### 3.2.1. Our Dataset

We developed the largest CBCT dataset for mandibular canal segmentation. To develop our dataset, 1010 dental CBCT scans were obtained from the PACS of the Seoul National University Dental Hospital. The data was annotated in two stages; in the first stage 28 trained medical students from Seoul National University Dental Hospital performed annotations and in the second stage 6 doctors from the same institute validated the annotated data. The CBCT scans were in DICOM format with voxel spacing ranging from 0.3 mm to 0.39 mm. The annotated data was available as a set of floating point polygon coordinates for each of the left and right canals, stored in JSON file format for every patient. The spatial resolution of scans ranges from 512 × 512 × 460 voxels to 670 × 670 × 640 voxels. The Field Of View (FOV) of all the scans in this data is large, as described in Figure 1. The large FOV captures the complete dentition, including both temporo-mandibular joints and the cranial base. The CBCT scans in our dataset consist of three Hounsfield Unit (HU) values ranges, i.e., from −1000 to +1000, −1000 to +2000 and 0 to 5000 HU as demonstrated in Figure 2. All the experiments are conducted using 100, 200, 300 and 400 scans for training and tested on 500 samples.

#### 3.2.2. Public Dataset

The 347 CBCT images in the public dataset [27] have a fixed pixel spacing of 0.3 mm and the Hounsfield Unit (HU) values for all scans fall within a fixed range of −1000 to 5264. The spatial resolution of scans ranges from 148×265×312 to 178×423×463. The 347 scans that make up this dataset are split into two parts; the primary dataset, which includes the 91 volumes for which both dense and sparse annotations are accessible and the secondary dataset, for which only the sparsely annotations are available. As shown in Figure 1b, the FOV of these scans are medium as opposed to our dataset which has large FOV.

Table 2 compares the two datasets utilized for experimentation and validation of our proposed technique. Figure 3 shows the 3D rendered scans from both datasets with mandibular canal annotations. In contrast to the public dataset with medium FOV, scans in our dataset are acquired with large FOV.

### 3.3. Data Pre-Processing

Although CT scans follow a worldwide standard for ranges of HU values for different body parts such as teeth, gums, bones, etc., 3D CBCT scans, follow no such standard and therefore can have different ranges of HU values and relative intensities when acquired from different manufacturers and under different scanning conditions. Fixed window levels and window widths can give varying contrasts and can be a cause for poor results during processing. To make the algorithm robust, dynamic windowing was applied to bring the contrast of all the scans to a similar level. This was done by calculating the window levels and widths (WL/WW) on run-time for each scan by analyzing the trend of the intensity histogram of the scan. This ensured a standard contrast of the scan after windowing. The intensity histogram of the three different types of CBCT scans, each acquired from a different manufacturer and the placement of their calculated Window Centre (WC) through the above logic, is shown in Figure 2. To perform the dynamic windowing, the intensity histogram of the individual scan was evaluated and the intensity with the highest frequency was set as the window center. The Window Width (WW) is determined by the range of intensities—the longer the range, the higher the window width—with less change in WW as the range reaches high intensities.

### 3.4. Overview of Dual-Stage Framework

The major problem faced in the segmentation of the mandibular canal is the imbalance between the mandibular canal and background classes. CBCT scans include the whole face and jaws, while the region of interest for mandibular canal extraction is only the jaw region. This problem often leads to misclassifications, especially at pixels on the boundary of the canal. Another problem that arises while refining the results of segmentation is computational power. Thus, in this study, our aim was to resolve these issues by using two cascade networks to produce a full-resolution segmentation output. The first CNN performed a coarse segmentation of the MC and the second network utilized the VOIs from the first network to produce refined segmentation. The output of the first model was used to isolate the left and right parts of the face as well as crop the regions around the mandibular canal. Hence, the model gives two VOIs, i.e., the region around the left mandibular canal and the region around the right mandibular canal. These cropped VOIs were used as the input of the second model, which produced the final fine segmentation of left and right mandibular canals.

#### 3.4.1. Jaw Localization

Jaw localization will inherently improve the accuracy and reduce noisy segmentation maps of the canal. Moreover, optimizing a model for full-size 1010 scans of dimensions 128×128×128 is computationally expensive as compared to a localized scan. Reducing the size of the scan would also affect the appearance of the mandibular canal. Hence, we utilized 20% of the data, i.e., 200 scans to train a localization model to coarsely segment the mandibular canal. This segmentation is used to roughly localize the canal and as an input to the second model for segmentation.

Since the anatomical contexts in 3D medical pictures are far more complex than those in 2D images, 3D variations of UNet with significantly more parameters are often needed to capture more representative characteristics. However, a large number of parameter weights and depth in the 3D UNet creates various optimization challenges, like over-fitting, slow convergence rate, gradient vanishing [31] and repetitive computation while training. However, the 3D CBCT scans contain much redundant information which significantly increases the network parameters and its optimization time.

In this study, these issues are resolved using Deeply Supervised Attention UNet architecture [32]. The input to the network is a 3D CBCT scan $x\in R^{128\times 128\times 128}$ and the output is a segmentation map $\Phi \left(x\right)\in {\left[0,1\right]}^{128\times 128\times 128}$. The model’s output is a segmentation mask that coarsely segments the canals.

The network consists of encoder and decoder blocks. The encoder network learns to extract the necessary information from an input image, which is then passed on to the decoder. Each decoder block consists of attention gate skip connections from the encoder. The attention gate assists the model in selecting more useful features. It takes two inputs: the up-sampling feature in the decoder and the corresponding depth feature in the encoder, as shown in Figure 4a. The feature from the encoder is used as a gating signal to enhance the learning of the feature in the decoder. Attention gates automatically learn to focus on target structures without additional supervision. At test time, these gates generate soft region proposals implicitly on runtime and highlight salient features useful for a specific task. Moreover, they reduce the computational load and improve the model’s sensitivity and accuracy for dense label predictions by suppressing feature activations in irrelevant regions.

In order to capture the inter-slice connectivity of the canal and obtain fine-tuned segmentation results, the framework combines the current 3D Attention UNet model with a 3D deep supervision mechanism during training. This strengthens the propagation of gradient flow inside the network and therefore acquires more effective and representative features. The 3D deep supervision method greatly regulates the training of the hidden layers. It is only used in training mode as it helps with segmentation by properly regularizing the network weights.

The segmentation masks produced in this step were used to divide the scan into left and right parts and the VOI for fine segmentation of the canal was extracted by cropping the region around the segmented canal.

#### 3.4.2. 3D Mandibular Canal Segmentation

The size of the mandibular canal was analyzed statistically, as shown in the Figure 5. It is visible that both the right and left canals vary in size. Thus, to ensure that the model’s performance remains similar for all sizes of canal VOIs, Residual UNet architecture with multi-scale inputs was utilized to perform the task of 3D segmentation of the mandibular canal, as shown in (b) in Figure 4. The sizes of three inputs are kept as 208 × 240 × 240, 144 × 176 × 176 and 108 × 144 × 144. The benefit of multi-scale input is that it caters to all the different available sizes of the mandibular canal and hence reduces the segmentation error around the boundary. This structure enables the encoder of the network to extract features better. The output was binarized by applying a thresholding of 0.5.

The ResUNet or Deep Residual UNet architecture was utilized for 3D mandibular canal segmentation [33], an architecture that relies on deep residual learning and UNet. Its structure can be divided into an encoding network and a decoding network. The two consecutive layers are applied to the basic residual block and the same padding is used in the encoding branch. A batch normalization layer follows each convolutional layer, followed by a ReLU layer (non-linear layer). Downsampling is done by max-pooling operation after the residual block. The number of feature channels is doubled at each down-sampling step. In order to restore the size of the segmented output, the same amount of up-sampling operations are carried out in the decoding network. A transposed convolution is used to achieve each up-sampling and the number of channels of feature is reduced by half. After passing through the channel attention block, skip connections are created to transfer features from the encoder to the decoder and basic residual blocks with two successive convolutional layers (with the same padding) are used for feature extraction. Similar to this, a batch normalization layer and a ReLU layer is placed after every convolutional layer. By using concatenation, which is employed in UNet, the encoded and decoded data are combined. The final segmentation masks, obtained from the network, are further refined by classical image processing techniques such as dilation, erosion and opening and closing morphological operations [34] to remove the noise as well as discontinuities in the canal.

### 3.5. Implementation Details and Training Strategy

For this study, all the CNN architectures were implemented using the Keras framework [35] with TensorFlow [36] as back-end. We performed our experiment on two powerful NVIDIA Titan RTX GPUs with 4608 CUDA cores and 24 GB GDDR6 SDRAM. The batch size for this experiment was set to 2 for both models and the proposed architecture was optimized with the Adam optimizer. The learning rate to train the model was set to 1 × 10^−5^. To reduce the training time and use the GPU efficiently, we use 10 to 20 percent of the 3D CBCT scans for training of model for jaw localization as well as canal segmentation. The size of images while training the jaw localization model is kept to 128 × 128 × 128. After localizing the jaw, the 3D images are cropped and resized to three fixed sizes as mentioned in Section 3.4.2. We used the dice loss function Equation (Equation 1) to calculate the loss. Labels are the segmentation annotation of images containing 0 as background and 1 as foreground. We trained the localization model for 50 epochs and the 3D segmentation models for 80 epochs by keeping the learning rate lower in order to train a generalized model. The batch size, epoch and learning rate were reset depending upon the need.
(1)$$\begin{array}{cc} \;\mathrm{Dice}\;\mathrm{Loss} & =1-\mathrm{Dice}\;\mathrm{Coefficient} \end{array}$$
where, dice coefficient is given by Equation (Equation 6).

### 3.6. Performance Measures

In order to measure the performance of the deep learning model, we calculated the dice score, mean IoU, precision, recall, *F*1 score and specificity using the following equations:(2)$$\begin{array}{cc} \;\mathrm{precision}\; & =\frac{TP}{TP+FP} \end{array}$$
(3)$$\begin{array}{cc} \;\mathrm{recall}\; & =\frac{TP}{TP+FN} \end{array}$$
(4)$$\begin{array}{cc} F1\;\mathrm{score} & =\frac{TP}{TP+0.5(FP+FN)} \end{array}$$
(5)$$\begin{array}{cc} IoU & =\frac{TP}{TP+FP+FN} \end{array}$$
(6)$$\mathrm{DSC}=\frac{2TP}{2TP+FP+FN}$$
where, TP refers to true Positives, FP refers to False Positives, FN refers to False Negatives and TN refers to True Negatives. IoU refers to Intersection Over Union and DSC is Dice Coefficient.

## 4. Results and Discussion

To evaluate our proposed framework, we performed various experiments on our dataset, the largest full-view CBCT dataset with voxel-level annotations of the mandibular canal. Additionally, we also tested on the public dataset [27], which has 91 medium view CBCT scans with dense annotations of the mandibular canal. We start our analysis by benchmarking the performance of dual-stage deeply supervised attention-based Convolutional Neural Networks over its other variants. Later, the results are quantitatively analyzed in terms of overall performance and qualitative/visual analysis.

### 4.1. Benchmarking Results

First, we benchmark the performance of our proposed framework with its possible modifications. Our framework consists of two networks, one for the localization and a second for the fine-tuning of mandibular canal segmentation. Among these, the second stage, on which 3D Multi-Scale input Residual UNet (MSiR-UNet) architecture is employed, is more critical as it performs the final segmentation of the mandibular canal. Therefore, we implemented two versions of our MSiR-UNet to demonstrate the contribution of each component. In particular, we implemented one network without multi-scale input and another without residual connections while keeping the remaining pipeline the same as the proposed framework. We trained all three models on 300 CBCT scans and evaluated their performances with respect to the evaluation parameters described in Section 3.6. Table 3 summarizes the results obtained from each model, clearly showing the contribution of each component of the proposed network.

Without multi-scale input, the model has to rely on a single dimension that comes from the resizing module, either after up-sampling or down-sampling of the original VOI. This limits the model to having untampered information which originally belongs to the sub-volume of the CBCT scan. Subsequently, the version without the incorporation of multi-scale input showed degraded performance. Similarly, the residual connections enable the flow of information from features of various scales in the network. Therefore, the version implemented without residual connection failed to achieve the same performance as the proposed MSiR-UNet. To this end, it can be safely concluded from our experiments that including multi-scale inputs and residual connections improve the network’s learning ability, enabling the network to achieve better segmentation results.

### 4.2. Impact of Increasing the Amount of Data

Although we developed a dataset consisting of 1010 CBCT scans and voxel level annotations of the mandibular canal, we trained our proposed dual-stage deeply supervised attention-based Convolutional Neural Network on different amounts of data to demonstrate the effectiveness of the presented framework. In particular, we trained three models on 100, 200, 300 and 400 CBCT scans and used the fixed 500 CBCT scans for testing. Table 4 summarizes the average results obtained on the test set with various evaluation parameters defined in Section 3.6. We observed that the model improves quite strongly initially when the amount of data is increased up to 300 scans; however, beyond that, we observe a plateauing effect in the performance. Concretely, the performance in terms of F1 score is improved by 1.6 % and 2.3 % for the left canal when we increase the samples from 100 to 200 and 200 to 300, respectively. However, only a 0.2 % increase in F1 score is observed when increasing the number of training samples from 300 to 400, which is not an optimal choice with respect to computational time. Therefore, for our final solution, we selected the model trained on 300 scans to obtain final results.

### 4.3. Overall Performance Analysis

We analyzed the overall performance of the proposed framework quantitatively while also comparing the results with previously published techniques with respect to the evaluation parameters described in Section 3.6. In the literature, most studies have utilized private datasets, which vary in terms of the number of samples and type of scan, i.e., full view or medium view. Therefore, in Table 5, we summarize the results obtained by existing deep learning-based methods which utilize private datasets, along with their dataset details and the nature of the solution for a fair comparison. In the listed techniques, Verhelst et al. [25] achieved the highest performance in terms of dice score; however, their method requires extensive human assistance, which makes it a semi-automated technique. Jakarta et al. [20], Kwak et al. [19] and Dhar et al. [24] used scans with medium FOV to develop a fully automatic solution for mandibular canal segmentation. Although Kwak et al. [19] utilized less data for training reported and reported the highest mIoU score, the work provides no information about the number of test samples and uses only one evaluation parameter, which is insufficient to prove the effectiveness of their method. On the other hand, Jaskari et al. [20] utilized a significantly large amount of training data and reported their performance in terms of dice score and F1 score, which is inadequate for clinic applications. Dhar et al. [24] used less data for training and testing their solution; however, they reported a similar performance as in [20]. Among all the studies, only Lahoud et al. [26] utilized large-view CBCT scans along with medium-view scans and achieved slightly better performance in terms of dice score. However, Lahoud et al. [26] used only 30 CBCT scans for testing, including large and medium FOV scans, while in this study, results are obtained on 500 large-view CBCT scans. It is important to mention that mandibular canal segmentation becomes more challenging in large FOV CBCT scans due to bigger volume and dimensions of scans. Nevertheless, the proposed framework achieved consistent performance on 500 scans, depicting its robustness against the variations caused by the scanning devices and facial structures.

We also evaluate our framework on a publicly available dataset [27] to demonstrate its effectiveness. We trained an independent model from scratch on the public dataset, following the same pipeline as described in Section 3.4. Concretely, we first trained deeply supervised UNet architecture for jaw localization and later trained multi-scale input residual UNet architecture for fine segmentation of the mandibular canal. We only used densely annotated scans for training. The data distribution was 76 scans for training and 15 for testing, which is the same distribution as used in [28]. Table 6 summarizes the results from our study and two previously published studies. Our method achieved an improved performance compared to previous baseline results obtained with only dense annotations. However, our overall performance on the public dataset is degraded compared to the performances achieved on our dataset. This can be attributed to the amount of data utilized for training the model with the public dataset, which is several times smaller than our dataset.

### 4.4. Qualitative Analysis

We further qualitatively analyze the proposed technique on our developed dataset and public dataset [27]. Firstly, as shown in Figure 6, we analyzed the results in a 2D manner using parasagittal, Maximum Intensity Projection (MIP) images of the front, left and right sides. It can be observed that the proposed framework can accurately track the mandibular canal curve and produces performance quite similar to the ground truth. Although the predicted results are smoother than the ground truth, such smoothness has no impact on the clinical application.

We extend our visual analysis to 3D by rendering the medium view scans from the public dataset [27]. Figure 7 shows the results obtained from the proposed framework along with the ground truth dense annotations. From visual analysis, it can be observed that our framework produces results quite similar to ground truth annotations. Despite achieving the clinically acceptable dice score of 0.75 [26], our results do not fully match with the annotations made by expert dentists. However, it is important to note that an expert takes almost an hour to segment the mandibular canal in a large view CBCT scan while our framework only takes less than a minute, demonstrating its effectiveness.

## 5. Conclusions

This paper proposes a novel dual-stage deep learning-based scheme for automatic detection of the mandibular canal. We first enhanced the CBCT scans by employing a novel histogram-based dynamic windowing scheme, which improves the visibility of mandibular canals. After enhancement, we designed a 3D deeply supervised attention UNet architecture for localizing the mandibular canals within the volumes of interest (VOIs). Finally, each VOI is fed to Multi-Scale input Residual UNet (MSiR-UNet) architecture to segment the mandibular canals accurately. The proposed method has been rigorously evaluated on our dataset as well as on the public dataset. An extensive quantitative and qualitative/visual analysis has been performed. The results demonstrate that our framework performs consistently on scans with different types of field of view, i.e., medium and large views. Furthermore, our framework achieves clinically acceptable performance on both datasets, which makes it suitable for real-time clinical application. Future work includes reducing the number of stages of our approach as this method requires training multiple models, leading to high computation costs and time.

## Figures and Tables

**Figure 1 sensors-22-09877-f001:**
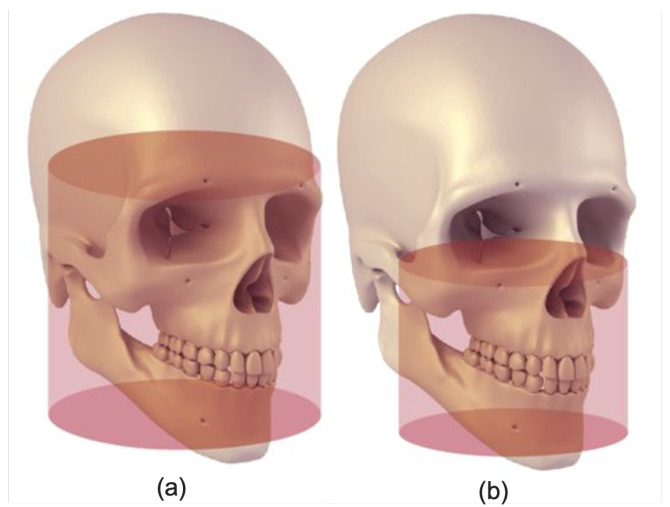
Field of views (FOVs) representing (**a**) Large FOV (140 mm × 165 mm) and (**b**) Medium (80 mm × 100 mm) (Figure Credit: [30]).

**Figure 2 sensors-22-09877-f002:**
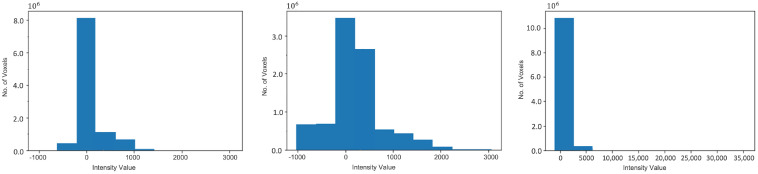
The intensity histograms of the different types of scans. WC represents the Window Centre calculated at run-time based on each histogram.

**Figure 3 sensors-22-09877-f003:**
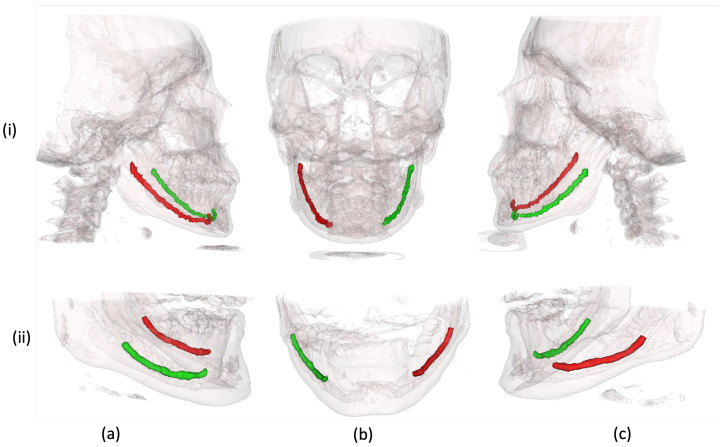
Three-dimensional rendered CBCT scans with left and right mandibular canal annotations. (**i**) Samples from our dataset. (**ii**) Samples from public dataset [27]. (**a**–**c**) the right, frontal and left side views of CBCT volumes, respectively.

**Figure 4 sensors-22-09877-f004:**
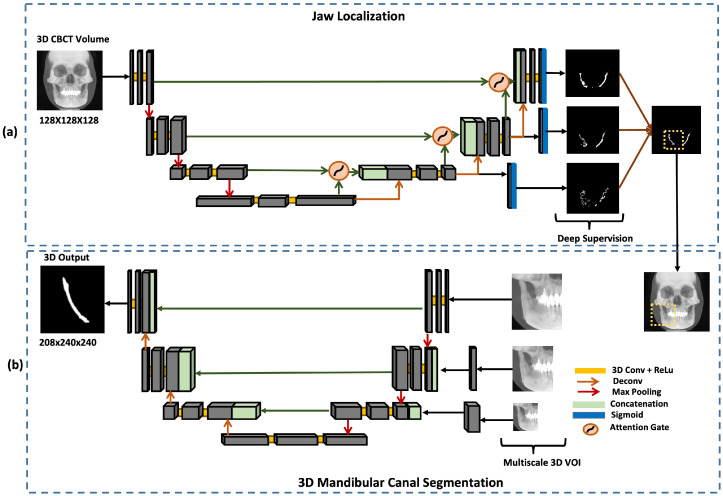
Proposed dual-stage scheme for mandibular canal segmentation, describing the model architectures utilized at each stage. (**a**) Deeply supervised attention UNet model for jaw localization which coarsely segments the canal. (**b**) Multi-scale input ResUNet model used to produce fine segmentation of mandibular canals (i.e., left and right canals).

**Figure 5 sensors-22-09877-f005:**
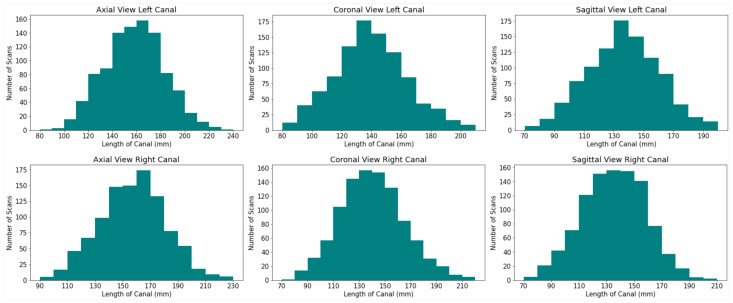
Histograms depicting the difference in sizes of the left and right mandibular canal.

**Figure 6 sensors-22-09877-f006:**
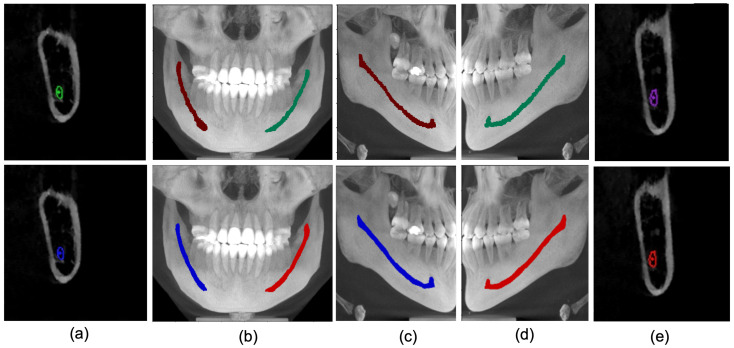
Ground truth (row 1) vs. Model Prediction (row 2). (**a**) Parasagittal view of right MC. (**b**) Maximum Intensity Projection of Coronal view (blue as right canal and red as left canal. (**c**) Right canal comparison with ground truth. (**d**) Left canal Comparison with ground truth. (**e**) Parasagittal view of Left Canal.

**Figure 7 sensors-22-09877-f007:**
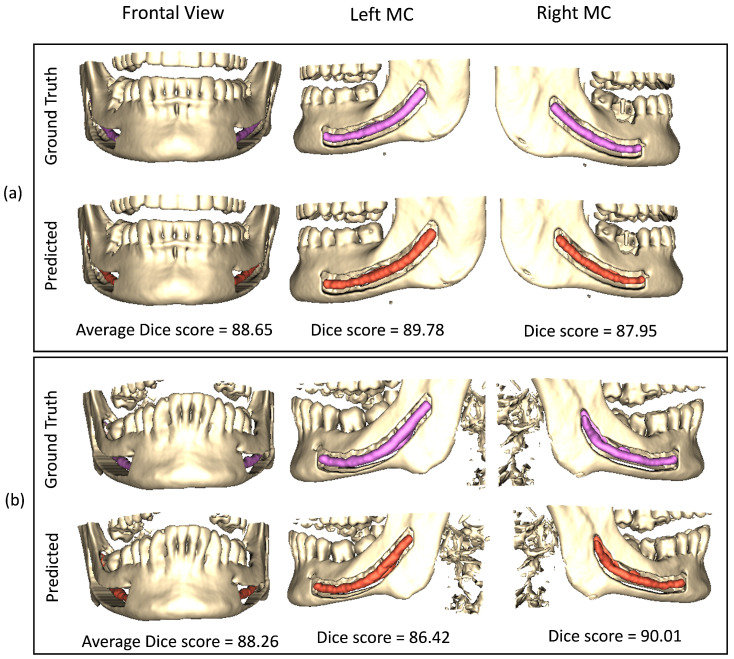
Visual results of our proposed model on public dataset on 3D rendered medium view CBCT scans. Sub-figures (**a**,**b**) show results from two different samples. The first and second row of each sub-figure represent the ground truth and predicted results, respectively.

**Table 1 sensors-22-09877-t001:** Summary of recently published deep learning-based methods for mandibular canal segmentation in CBCT scans.

Author, Study and Year of Publication	Technique	Type of Dataset and FOV	No. of CBCT Scans		Contributions	Limitations
			Training + Validation	Test		
Kwak et al. [19], 2020	Thresholding-based teeth segmentation + 3D UNets	Private, Full View	82	20	Employed 2D and 3D Deep Learning models and demonstrated the superior performance of 3D UNets	Limited performance in terms of Mean mIoU
Jaskari et al. [20], 2020	3D Fully Convolutional Neural Networks (FCNNs)	Private, Medium View	509	128	The study utilized a large number of CBCT scans to train 3D FCNNs and achieved an improved performance.	Overall achieved performance for left and right canal was limited in term of Dice score
Faradhilla et al. [21], 2021	Residual FCNNs + Dual auxilary Loss functions	Private, 2D view	NA	NA	The study exploited Residual Fully Convolutional Network with dual auxiliary loss functions to segment the mandibular canal in parasagittal 2D images and reported promising results in terms of dice score	Requires manual input from dentists to generate the 2D parasagittal views from CBCT. Study provides no information about the CBCT scans used for the experimentation
Verhelst et al. [25], 2021	3D UNet trained in two phases	Private, Medium View	160	30	Trained 3D UNet in two phases, i.e., before and after the deployment, to achieve promising performance.	Requires an extensive effort to train the model and inputs from experts are needed to improve its performance of the model.
Widiasri et al. [22], 2022	YOLOv4	Private, 2D view	NA	NA	The study utilized YOLOv4 for mandibular canal detection in 2D coronal images and achieved significantly higher detection performance.	The study used 2D coronal images which need manual input to generate from CBCT scans. The technique just provides the bounding box around the canal region, which lacks the exact boundary information which can be obtained from segmentation.
Lahoud et al. [26], 2022	Two 3D UNets, one for coarse segmentation and other for finetuning on patches	Private, Mixed FOVs	196	39	Adjusted to the variability in Mandibular Canal shape and width by using voxel-wise probability approach for segmentation	The scheme requires an extensive effort to train the models and evaluate performed on a limited private dataset does not prove the generalization
Cipriano et al. [27], 2022	Jaskari et al. [20], 2020	Public, Medium view	76 with Dense annotation	15 with Dense Annotations	The first publicly released annotated dataset and source code, validated their dataset on three different existing techniques	Utilized the existing segmentation methods, with no contribution in terms of technique novelty.
Cipriano et al. [28], 2022	3D CNN + Deep label propagation technique	Public, Medium view	76 with Dense annotation+256 with Sparse Annotations	15 with Dense Annotations	Combined 3D segmentation model trained on the 3D annotated data and label propagation model to improve the mandibular canal segmentation performance	The study utilized the scans with a medium Field Of View (FOV) which is 3D sub-volume from CBCT scans, however, no mechanism for localization of medium FOV is provided.

**Table 2 sensors-22-09877-t002:** Comparison of our dataset with public dataset.

	Our Dataset	Public Dataset [27]
Total Number of CBCT scans	1010	347
Densely annotated scans	1010	91
Sparsely Annotated scans	-	256
Minimum Resolution	512 × 512 × 460	148 × 265 × 312
Maximum Resolution	670 × 670 × 640	178 × 423 × 463
Pixel Spacing	0.3 mm–0.39 mm	0.3 mm
Field of View	Large	Medium

**Table 3 sensors-22-09877-t003:** Benchmarking results of the proposed Multi-Scale input Residual UNet (MSiR-UNet) architecture against its two versions, i.e., without multi-scale input and residual connections, by using various evaluation parameters.

Performance Parameters	Without Multi-Scale	Without Resiudual Connections	With Residual Connections and Multi-Scale Inputs
**mIoU**	0.779	0.785	0.795
**Precision**	0.683	0.679	0.69
**Recall**	0.81	0.824	0.83
**Dice Score**	0.72	0.72	0.751
**F1 Score**	0.741	0.745	0.759

**Table 4 sensors-22-09877-t004:** Impact of increasing the amount of data on the performance of model.

Number of Scans	100		200		300		400	
Canal Side	Left	Right	Left	Right	Left	Right	Left	Right
mIoU	0.755	0.771	0.78	0.789	0.79	0.8	0.798	0.806
Precision	0.639	0.667	0.657	0.69	0.679	0.718	0.686	0.72
Recall	0.818	0.795	0.832	0.8	0.847	0.817	0.854	0.819
Dice Score	0.721	0.731	0.734	0.746	0.749	0.753	0.752	0.759
F1 Score	0.718	0.725	0.734	0.741	0.754	0.764	0.761	0.766

**Table 5 sensors-22-09877-t005:** Comparison of our technique with other techniques on private dataset.

Study	Field of View	Training Scans	Testing Scans	Solution Type	mIoU	Precision	Recall	Dice Score	F1 Score
Jaskari et al. [20], 2020	Medium	457	128	Automated	-	-	-	0.575	-
Kwak et al. [19], 2020 (3D)	Medium	61	-	Automated	0.577	-	-	-	-
Dhar et al. [24], 2021	Medium	157	30	Automated	0.7	0.63	0.51	-	0.56
Verhelst et al. [25], 2021	Medium	196	39	Semi-Automated	0.946	0.952	0.993	0.972	-
Lahoud et al. [26], 2022	Medium + Large	166	39	Automated	0.636	0.782	0.792	0.774	-
		100	500	Automated	0.763	0.653	0.807	0.726	0.721
Our method	Large	200	500	Automated	0.785	0.67	0.816	0.74	0.737
		300	500	Automated	0.795	0.69	0.832	0.751	0.759

**Table 6 sensors-22-09877-t006:** Results of our technique on public dataset.

Study	IoU	Dice Score
Jaskari et al. [20], 2022	0.52	0.67
Cipriano et al. [28], 2022	0.61	0.75
Our method	0.79	0.77

## Data Availability

The dataset developed in this study was under artifical intelligence training data collection project which was funded by National Information Society Agency (NIA), South Korea. The dataset has been made available on this link and can be accessed after getting approval from NIA. https://www.aihub.or.kr/aihubdata/data/view.do?currMenu=115&topMenu=100&aihubDataSe=realm&dataSetSn=203 (14 December 2022).

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
