# Peer review of "Dual-Stage Deeply Supervised Attention-Based Convolutional Neural Networks for Mandibular Canal Segmentation in CBCT Scans"

_sensors, 2022, doi:10.3390/s22249877_

Round 1
Reviewer 1 Report
The paper presents a deep learning based image segmentation approach for CBCT scans. The paper belongs to an interesting research area, however there serious concerns about the quality, rigour, novelty and presentation of the reported research findings.
The following concerns can help the authors to significantly improve the paper quality for a re-submission to this journal or other avenues might be considered.
1. Plagiarism issue: The paper is already available at the following locations with almost the same content. Please check with MDPI Editorial office if this is allowed or will it be considered as plagiarism.
https://deepai.org/publication/dual-stage-deeply-supervised-attention-based-convolutional-neural-networks-for-mandibular-canal-segmentation-in-cbct-scans
https://arxiv.org/abs/2210.03739
2. English language issue: There are multiple places in the paper where grammatical (line 3, 145, 238), punctuation (line 2), capitalisation (line 21, 37) and incomplete sentence structures (line 31) have been found. Please make sure that a native English reader proofreads the manuscript before submitting it again.
3. Flow issues: The flow of the paper requires lot of improvement. The order in which the results have been presented is improperly planned (e.g. Table 1, Then Table 3 and then Table 2). Please re-order for better readability and understanding to the reader.
4. Related work: It is recommended that a separate "Related Work" section is included in the paper (lines 53 to 73). Also, the related work should provide a critical evaluation of the existing research. It is better to summarize this in the form of a comparative table (with approaches used, dataset utilized, pros, cons) which will help to identify the research gap this paper is addressing.
5. Dataset concern: Please use standard datasets in your study or make your dataset publicly available. This will enable other researchers to validate your study results. Also, the number of images for the dataset (1000) is small when it comes to Deep Learning applications.
6. Figures issue: The figures are not clear (Fig. 1, 3) and small in size which makes them unreadable. The location of the figures are not in sync with their textual explanation (e.g. Fig 1 is on page 2, while its explanation is on page 5). Significance of Fig 2 results in the study is not clear. Figure 5 requires more contrast for better visuals.
7. Text and Figure mismatch: Is the text at lines 185-200 for Fig. 1 or for another Deep Learning architecture?
8. Table issues: Table 3 and Table 4 are not presented in a complete manner. Some of the columns are not displayed as they are too wide.
9. Incomplete citations: Line 202, the citation number is missing (shows as a ? mark)
10. Results and Discussion: The results and their discussion is not obvious and the reader is unclear about the conclusions obtained (e.g. line 223).
11. Comparative study: Please improve the comparative study section with a most relevant and recent study on a similar dataset. Other wise comparison does not hold valid.
12. References: The number of references should be increased. More recent references are recommended from 2021/2022. Most of the references (from 1 to 8) are from the medical field (dentistry). It is advised to include more paper from the domain of Deep Learning and the medical field combined. Ref [15] looks out of context (why a Geoscience paper is cited?)
13. Minor issues: Section names are inconsistent (Introduction is called as Background & Related work, in line 74). In line 157, Figure 1 is called as Diagram 1. The model name has different versions (U-Net in line 139, UNet in line 148)
Author Response
We would like to firstly thank the reviewers for their valuable comments, which helped us in improving the quality of paper in several places. We incorporated each comment into consideration and did the necessary modifications in the revised version.
In the attached document, we provide a summary of all the modifications and point-by-point reply to the reviewer recommendations. To improve the readability of this document, and to help differentiate the reviewer’s comments from our response, we have used different color formatting in this document.

Reviewer 2 Report
The authors present a framework to extract mandibular canal. The accuracy is comparable with other study. Several comments:
1. Verhelst et al. 2021 has extremely high precision and recall compared with others and this paper. What is special?
2. what is a acceptable dice score for clinical use?
3. Tables are out of page.
Author Response
We would like to first thank the reviewers for their valuable comments, which helped us in improving the quality of the paper in several places. We incorporated each comment into consideration and did the necessary modifications in the revised version.
In the attached document, we provide a summary of all the modifications and a point-by-point reply to the reviewer's recommendations. To improve the readability of this document, and to help differentiate the reviewer’s comments from our response, we have used different color formatting in this document.

Round 2
Reviewer 1 Report
There has been a significant improvement in the paper, as most of the comments have been incorporated.
However, it is still not easily identifiable what exact changes have been made with respect to the following comments.
It is advised that the authors clearly highlight (with red color and mentioning the line numbers) these specific changes in the v2 of the manuscript.
2. English language issue: There are multiple places in the paper where grammatical (line 3, 145, 238), punctuation (line 2), capitalisation (line 21, 37) and incomplete sentence structures (line 31) have been found. Please make sure that a native English reader proofreads the manuscript before submitting it again.
12. References: The number of references should be increased. More recent references are recommended from 2021/2022. Most of the references (from 1 to 8) are from the medical field (dentistry). It is advised to include more paper from the domain of Deep Learning and the medical field combined. Ref [15] looks out of context (why a Geoscience paper is cited?)
13. Minor issues: Section names are inconsistent (Introduction is called as Background & Related work, in line 74). In line 157, Figure 1 is called as Diagram 1. The model name has different versions (U-Net in line 139, UNet in line 148)
Author Response
We would like to thank the reviewer for his/her valuable feedback, which helped us improve the quality of the paper in several places. We strived to incorporate each comment into consideration and made the necessary modifications in the revised version. However, we regret that the reviewer faced inconvenience while tracking the changes in the valuable comments (i.e., 2, 12, and 13).
In the attached document, we provide a summary of all the modifications and a point-by-point reply to the reviewer's recommendations. We have used different color formatting to improve this document's readability and help differentiate the reviewer’s comments from our response.

Reviewer 2 Report
The authors have answered my questions
Author Response
We would like to thank the reviewers for their thoughtful comments and efforts towards improving our manuscript.
